# System Dynamics Modeling in Additive Manufacturing Supply Chain Management

**Jairo Nuñez Rodriguez [1,\*], Hugo Hernando Andrade Sosa [2], Sylvia Maria Villarreal Archila [3] and Angel Ortiz [4]**

1 Faculty of Industrial Engineering, Universidad Pontificia Bolivariana, Km 7 Vía Piedecuesta, Piedecuesta 50031, Colombia

2 Research Group SIMON Universidad Industrial de Santander, Calle 9 con Carrera 27, Bucaramanga 680006, Colombia; handrade@uis.edu.co

3 Faculty of Industrial Engineering Unidades Tecnológicas de Santander, Avenida los Estudiantes Número 982, Bucaramanga 680005318, Colombia; sy.villareal@correo.uts.edu.co

4 Research Centre on Production Management and Engineering, Universitat Politècnica de València, Camino de vera s/n, 46022 Valencia, Spain; aortiz@cigip.upv.es

\* Correspondence: jairo.nunez@upb.edu.co; Tel.: +57-6796220

**Abstract:** A system dynamics model was developed with the primary purpose of visualizing the behavior of a supply chain (SC) when it adopts a disruptive technology such as additive manufacturing (AM). The model proposed a dynamic hypothesis that defines the following issue: what is the impact of the AM characteristics and processes in the SC? The model was represented through a causal diagram in thirteen variables related to the SC, organized in two feedback cycles and a data flow diagram, based mainly on the three-essential links of the SC and the order display traceability: supplier–focal manufacturer–distribution Network. Once proposed, the model was validated through the evaluation of extreme conditions and sensitivity analysis. As a result, the dynamic behavior of the variables that condition the chain management was analyzed, evidencing reduction times in production, especially in products that require greater complexity and detail, as well as reductions in inventories and the amount of raw material due to production and storing practices from AM. This model is the starting point for alternative supply chain scenarios through structural operating policies and operating policies in terms of process management.

**Keywords:** supply chain (SC); additive manufacturing (AM); system dynamics (SD)

## 1. Introduction

The application of modeling with system dynamics (SD) in supply chain management (SCM) has its roots in Forrester's Industrial Dynamics, through the definition of a model structured in six flow systems that interact among them transmission of the information, materials, orders, money, labor, and capital [1]. Forrester developed one of the first related models, in which he considered the chain as part of an industrial system in terms of policy design [2]. Nowadays, the model has evolved towards the concept of a testing environment for business management systems to make the right decisions regarding strategies and variable changes [3]. However, current trends and the transition to industry 4.0 and technological impact issues require support from system dynamics to respond to the behavior of variables influenced by breakthroughs, such as big data, cybersecurity, augmented reality, cloud computing, the Internet of Things (IoT), and additive manufacturing in SC management. Understanding them as value chain networks, where the resources and activities necessary to create and deliver services/products are involved to satisfy the customer's needs [4].

Regardless of the industry where they are analyzed, the supply chains are still considered complex and dynamic systems that involve different participants (stakeholders). These SCs contemplate internal and external factors, the importance of which increase or

decrease according to the type of product or service generated. In this sense, SCM is now regarded as the core of the companies' strategy mainly because the business competition at the moment is entirely chain-based, which requires an articulation of all the elements that make up the links [5]. Therefore, the Global Supply Chain Forum (GSCF) identified the eight standard processes that define chain management [6], of which relationship management with customers and suppliers is the element that constitutes the external link of the focal manufacturer, allowing a general view (from end to end). In comparison, the remaining six processes are specific to the internal processes and dynamics of the entity being analyzed [6].

The "new" manufacturing process, additive manufacturing (AM), which is capable of creating objects from 3D data, generally by adding layer upon layer [7], has captured the interest of academia and industry due to the exponential growth over the past 30 years since its inception. Among the potential functions attributed to it, it is expected to transform supply chains (SC) towards more localized approaches [8], which would generate changes in the replacement techniques, production costs, packaging, labeling, and product storage [9]. In this sense, the SC will suffer some critical changes in its elements and steps (components, structure, and processes), which have not yet been explored in depth, given the emerging state of the application of AM in the processes of different industries.

Given that system dynamics thinking allows modeling and understanding the SC behavior and the changes in variables over time, this paper's objective is to propose an SD model to understand and observe the management processes dynamics, considering all the variables that AM could potentially influence.

In order to elaborate a system dynamics model that would explain supply chain behavior with additive manufacturing, the authors consulted research related to SD and SC that has been published and documented for scientific literature to identify the necessary variables. It was possible to notice a time window from 1993 to 2019 where literature review and application studies were classified. It was possible to determine the applications of system dynamics in SC variables and behavior, highlighting the trend towards the appropriation of disruptive Industry 4.0 technologies. Subsequently, the stages of Sterman's process were applied with the guidance of Andrade [10], which include:

1.  Conceptualization of the model where the context and components of the supply chain are defined.
2.  The proposal of the diagram of influences, which includes a dynamic hypothesis that facilitates understanding the behavior of the interaction of the variables.
3.  The design of the data flow diagram.
4.  The mathematical formulation that includes the model equations.
5.  The validation of the model.
6.  The sensitivity analysis.

## 2. Literature Review

In order to learn about the applications of SD in SC modeling, a literature review was carried out to determine bibliometric indicators such as annual behavior, impact, and geographic concentration. The applications were characterized based on Lambert & Cooper's framework of key elements for chain management. Finally, researchers identified and classified the variables in the eight management processes defined by the Global Supply Chain Forum. The Web of Science and Scopus databases were consulted with the search equation: (TITLE (system AND dynamic*) AND TITLE (supply AND chain*)) that intercepted the two topics of interest: SD and SC, in a time window of 1993–2019 for a total analysis of 306 documents.

Complementarily, the ten papers with the highest impact were analyzed, which were published between 2000 and 2012. Half of the results correspond to proposals for system dynamics models: remanufacturing in closed-loop [11], control and management in construction [12], logistics [13], strategic food chain [14,15]; another three to the analysis

of dynamic systems: effectiveness of e-tools [16], sustainability [17], and recycling [18]; and the remaining two to reviews of model literature and chain systems [19,20].

In order to know the applications that SD has had in the analysis of SC management; it was necessary to carry out a keyword analysis through time. It was found that the first investigations focused on articulating the concepts through simulations and analysis of the Bullwhip effect. During the 1970s, there was an emphasis on analyzing variables such as reverse logistics, inventory control, supply chain system, optimization, stability, manufacturing, remanufactured products, and risk management, which are still in use today. By 1985, discrete event simulation was included in developing models and simulations, followed by evaluating variables related to green supply chains, sustainable development, and sustainability, which happened five years later.

Between 1990 and 2000, the paperwork related to Co-powersim (SD software) was associated with multi-agent systems and SC financial analysis. More recently, from 2008 to 2014, the study of these variables has continued, with variations in the evolution of concepts such as customer satisfaction, chain performance, uncertainty analysis, life cycle, and trade, among others, as shown in Figure 1. From 2014 to the present, the research has focused on modeling proposals that articulate collaborative models and disruptive technologies in the structure of the chain.

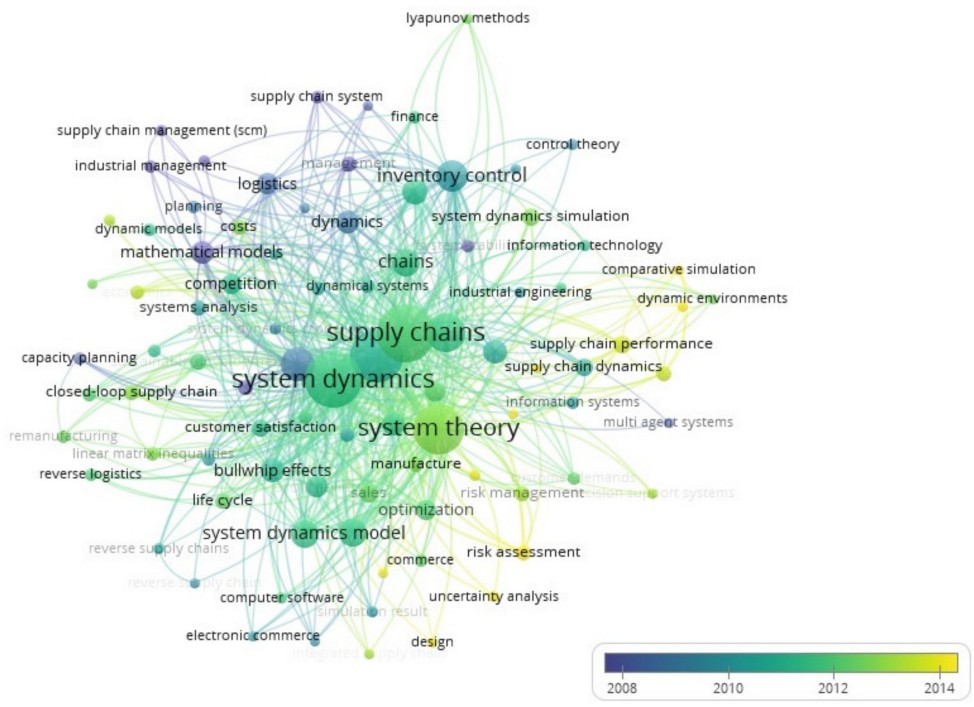

**Figure 1.** Keyword frequency analysis over time. Source: own elaboration based on results obtained by Vosviewer software.

The results show that SD initially focused on modeling internal variables of the chain (inventory management, responsiveness, manufacturing, and optimization). However, currently, it has been integrating external variables that impact the chain from the traditional approach (risk management, sustainability, collaboration models), which leads to including variables from the Industry 4.0 approach [21]. Consequently, it has been possible to identify different opportunities that can allow integrating digitalization trends through additive manufacturing, big data, and augmented reality, which will impact its configuration and performance [22]. Researchers have also identified the direction of efforts to include constructs that explain each of the trends (e.g., personalization and knowledge management), which are usually analyzed from a subjective or specific perspective for an application case, preventing a holistic analysis of the chain management. In response, multi-criteria models have been developed to delimit strategies for managing chain disruptions since

the application of these technologies requires a practical approach that mitigates risks and ensures sustainability [23].

The application results were grouped into the key elements and decisions from the supply chain model developed by Lambert and Cooper: processes, components, and network structure [24]. It was also found that 75% of the studies are related to the analysis of the variables from the supply chain management processes, in which the focus is to study one or two variables that model a specific process. There were coincidences in some elements associated with the model's mere application, such as processes and components or structure and processes (Supplementary Materials Table S1). In most application cases, a specific issue of a variable that affects a particular management process was modeled, which leads to understanding the variable behavior within the chain's performance. The predominant variables are inventory management, information transmission, chain integration, system performance, demand projection and management, and production planning and scheduling.

However, it was impossible to discover any reference where all the chain management processes were approached together from a holistic perspective. Although some of the paperwork did consider the chain as a system, where its variables affect each other, they did not integrate the eight SC management processes within a single model. Based on the literature review, the variables that influence and affect the SC management were identified, which are also categorized into the eight management processes defined by Lambert & Cooper from The Ohio State University (2000) [24] (Supplementary Materials Table S2). After analyzing the studies mentioned above, Table 1 presents the variables for the proposed model.

**Table 1.** Variables that affect SCM.

| Process | Variables |
|---|---|
| Customer relationship management | Cost effectiveness; service level; variability |
| Customer service management | Availability |
| Manufacturing flow management | Design cycle; minimum lot size; production capacity |
| Complete order management | Production order; delivery dates |
| Supplier relationship management | Orders, purchases, raw materials inventory, acquisition of materials |
| Demand management | Demand, inventories, variability |
| Product development and marketing | Design cycle |
| Returns management | Asset recovery; product availability |

Source: own elaboration.

In the customer relationship management process, structures are defined to develop and maintain loyal relationships with customers, where the product variability and service levels condition cost effectiveness. On the other hand, in the supplier relationship management process, the structure is defined to foster relationships with them, according to their long-term importance, modifying variables such as orders, raw material inventory, and acquisition of materials that ultimately translate into orders. Customer service management is the relationship between the company and the consumer. The company manages to identify essential information about the consumer and later on monitor the compliance on the product services agreements seeking to modify the availability variable. On the other hand, demand management is related to balancing customer requirements with supply chain capabilities, reducing variability, and increasing flexibility.

Complete order management is in charge of guaranteeing the integration of manufacturing, logistics, and marketing, making it easier to meet customer requirements and reduce delivery costs, which are conditioned by production orders and final dates. The manufacturing flow management includes all the activities necessary to manufacture the products requested by the demand, at the minimum cost, modifying the design cycle, the minimum batch size, and the production capacity. The product development marketing process corresponds to the essential activities to develop and bring products to market,

which is mainly evidenced in the design cycle. Finally, returns management helps entities to achieve a competitive advantage related to sustainability, identifying opportunities for improvement and innovation projects, translated into variables such as asset recovery and product availability.

In the literature there is research that reflects the behavior of supply chain management variables affected by the implementation of AM that generates: (+) personalization, less (−) residues, (−) energy consumption, (−) object weight, (−) cycle time, (−) associated tools, (−) transport cost, (−) environmental impact, (−) inventory, (−) number of people, (+) capital ratio, (+) geometric & design, (+) flexibility, (−) lots, (−) production, (−) waste, (−) chain links, (+) relationship with the consumer, (+) greater capital investment, (−) material, (+) collaborative relationships, and (−) product cost [25]. Similar research that has analyzed the impact of AM on CS was consulted, as presented in Table 2.

**Table 2.** Studies of Impacto f the additive manufacturing in the supply chain.

| Research | Methodology | Purpose |
|---|---|---|
| The impact of Additive Manufacturing on Supply Chain design: A simulation study [26] | Discrete event simulation model (Excel) | Quantitative evaluation of the effect of additive manufacturing on supply chain performance through system configuration. |
| Investigating the Impacts of Additive Manufacturing on Supply Chains [27] | Case study-surveys | To analyze the applications of AM in supply chains. This research focuses on the characteristics and traditional structure, which ends up designing an optimal business model to use. |
| The impact of 3D printing on manufacturer–retailer supply chains [28] | Mathematical model | To represent a simple supply chain consisting of a manufacturer and retailer that serves a stochastic customer demand that uses 3d printing to produce. |
| How will the diffusion of additive manufacturing impact the raw material supply chain process? [29] | System dynamics | To represent a model that represents the initial stage of the supply chain (raw material supply) by evaluating the reduction of materials inventories through the adoption of AM. |
| Additive manufacturing impacts on a two-level supply chain [30] | Joint Economic Lot Sizing model | To determine the impact of AM implementation in a two-level supply chain, focusing on inventory, transportation, and production costs. |
| Traditional vs. additive manufacturing supply chain configurations: A comparative case study [31] | Configuration theory, postulated by Alfred Chandler and widely applied in studies of TM and service SCs | To design a framework to determine the impacts on chain actors' operations by developing different modes and levels of products. |
| Impact of additive manufacturing on aircraft supply chain performance: A system dynamics approach [9] | System dynamics | It consists of evaluating the impact of AM implementation in a case study: aircraft supply chain. It was performed with theoretical data due to the absence of real-life data. |
| Topological network design of closed finite capacity supply chain networks [32] | Mathematical model | To analyze the layout, location, and arrangement of quotes for supply chains. |
| Additive manufacturing technology in spare parts supply chain: a comparative study [33] | System dynamics | To compare three supply chain scenarios and contrast the differences between costs and carbon emissions. |
| Additive manufacturing of biomedical implants: A feasibility assessment via supply-chain cost analysis [34] | Stochastic programming model | To determine the production costs of biomedical implants using AM. To determine the feasibility of manufacturing the implants on-site (hospitals). |
| Additive Manufacturing in an End-to-End Supply Chain Setting [35] | Optimization model | To determine the most critical factors to consider in the configuration stage of the SC that AM impacts. |
| Impact of additive manufacturing adoption on future of supply chains [36] | System dynamics | To describe the changes in the SC performance and structure as a result of the additive manufacturing implementation. To describe the characteristics and requirements of the chain. |
| The impact of additive manufacturing in the aircraft spare parts supply chain: Supply chain operation reference (SCOR) model-based analysis [37] | SCOR | To evaluate the impact of AM in the aircraft spares SC according to the SCOR operating model. |

Source: own elaboration.

In the studies mentioned above, the benefits of adopting AM in SCs were identified through a holistic analysis of chain behavior. It was possible to observe that 3D printing on supply chain management operations and relationships is relatively unexplored in the scientific literature, considering that studies have worked on changes in the chain's performance according to its structure in terms of time and costs. A particular focus has been given to mitigating inventory risks and the reduction of logistics operations. It was possible to identify that the literature lacks a better understanding of the nature of the changes generated by AM in chain management (processes), i.e., from a holistic perspective.

One of the methodologies most used by studies has been system dynamics [9,29,33,36], which started with Forrester [38] as a technique to mathematically model complex problems [20]. SD has become one of the most appropriate alternatives for analyzing supply chain management with structure and process policies. In addition, it has been widely used to represent the information and material flows [39] that are typical of SC analysis.

## 3. Materials and Methods

Researchers carried out a modeling process to the behavior of the variables related to the management of a supply chain driven by multi-product and multi-demand orders where the additive industry operates. The modeling was carried out through system dynamics, and its objective was to determine the additive industry's impact on the previously mentioned supply chain.

Researchers chose a methodology helpful when simulating supply chains through a modeling process such as system dynamics. Some of the advantages of a simulation carried out from that kind of modeling are:

- Discussion and understanding of complex issues.
- Creation and validation of scenarios that constitute the fundamental structure of a constantly changing system [40].
- Comprehension of the different relationships that might emerge between the system elements that are being analyzed.

During the model proposal, researchers followed Andrade's guidelines, which define the process as iterative, allowing them to review the stages as often as necessary. They also considered Sterman's process stages [10] that enable the validation of the proposal.

Figure 2 represents the language system in which the casualty from this model can be expressed: prose language and graphs with causal loop diagrams, flow and level diagrams, and equations as mathematical representation.

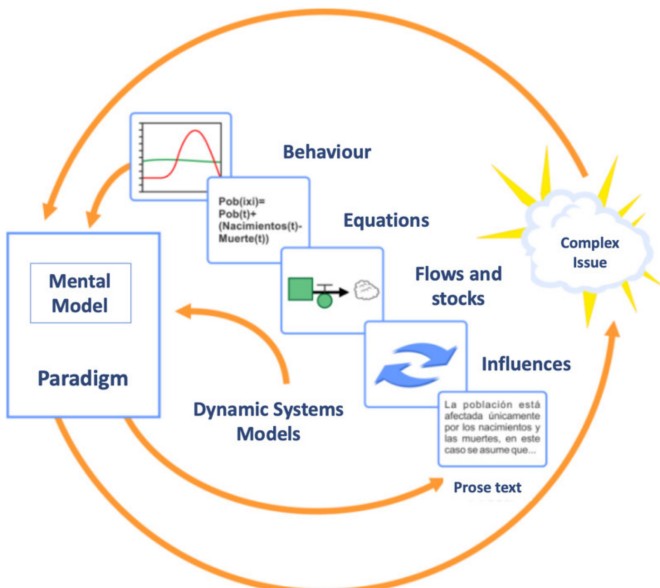

**Figure 2.** System Dynamics Modeling. Taken from: [40].

The results are structured in stages as follows:

- The conceptualization stage corresponds to the definition of the problem to be modeled.
- The variables of the model are adjusted to the conventions of the DS methodology.
- The influence diagram stage depicts the hypothesis' dynamic.
- The flow and level diagram with the proposed model is given.
- The mathematical formulation, meaning, and the equations are given.
- The model validation throughout tests of extreme conditions and sensitivity is given.

The model's output variables are related to the entire supply chain's cycle times and to the raw materials and finished goods inventory to contrast the subtractive manufacturing approach's demeanor and the additive manufacturing approach. The results were obtained by the Vensim and Evolution 4.6 software.

## 4. Results

### 4.1. Conceptualization

Figure 3 presents a supply chain's operating model made up of three links: supplier, manufacturer or assembler, and distribution network. It corresponds to a representation of a Make to Order (MTO) system; that is, production is only activated when the demand places an order. To define the structure, the characteristics that describe production with additive manufacturing, which are possible to find with traditional manufacturing, were reviewed.

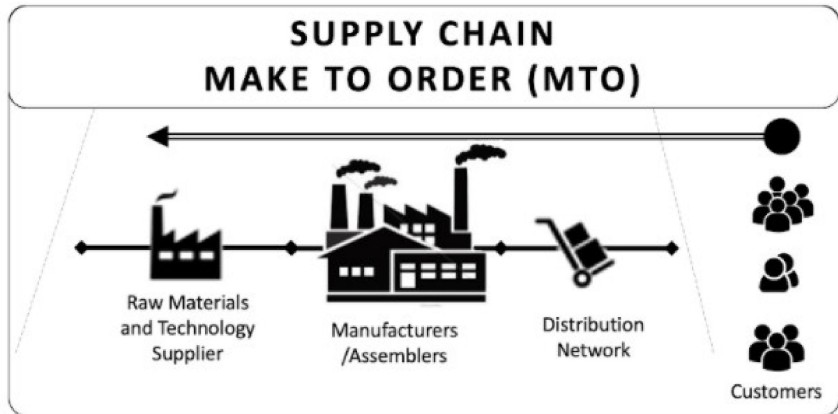

**Figure 3.** Operation model. Order-based supply chain (MTO). Source: own elaboration.

MTO modes respond to customer requests, resulting in increased customization opportunities with limited inventory time. Simultaneously, they generate an increase in the demand response time due to the acceptance capacity and the raw material inventory [41]. Likewise, they involve small production batches and variable raw material consumption and times. Consequently, inventory levels (WIP and finished goods) are kept to a minimum and, in many cases, even amount to zero. Theoretically, the response time is slow since companies want to finish all the activities before delivering the order.

When the order is received, the producer analyzes their production capacity and generates purchase orders for raw materials to start manufacturing. Considering that the inventory level tends to zero, suppliers are used, where variables such as response time, order quantity, and deliveries are analyzed.

Subsequently, the transformation cycle begins. Once the final product is obtained, the distribution network will deliver the product to the customers. Certain variables occur in this stage, such as delivery times, means of transportation, and the finished goods' quantity and size.

For the representation of the model through SD, researchers analyzed a multi-demand and multi-product assembly company [42]; with variable capacity and different characteristics for each product. The modeling includes suppliers of a single raw material, corresponding to an easily accessible commodity product that does not require a specialized supplier, and distributors with varying delivery times depending on the selected

means of transportation. These actors are articulated in the supply chain that functions as a collaborative system to respond to demand. This behavior can also be seen in a traditional supply chain, making it easier to contrast and quantify the manufacturing approach's impact on SC management.

The model is intended to represent additive manufacturing characteristics in order-based supply chain management to evaluate the behavior over time in terms of cycle time, raw material inventory, and finished goods inventory. The values worked in the model are discrete [43].

The main characteristics it represents are:

1. Multi-product with variable demand quantities;
2. Varied material consumption for each product;
3. Variable processing time for each product.

Based on these characteristics, the model seeks to answer questions such as these: What would happen if demand increases or decreases? What would happen if production capacity increases or decreases? What would happen if purchasing and distribution times change? What would happen if the policies of the assembly company or distribution network change?

### 4.2. Influence Diagram

Based on the context and the identification of variables, the dynamic hypothesis is formulated to facilitate understanding the behavior of the interaction of these variables. The diagram of influences presented in Figure 4 was constructed to propose the dynamic hypothesis. These diagrams represent the feedback mechanisms, which can be negative (balancing) or positive (reinforcement). In addition, they make two significant contributions to SD methodologies since they are preliminary sketches of the hypotheses and simplify the model proposal [11].

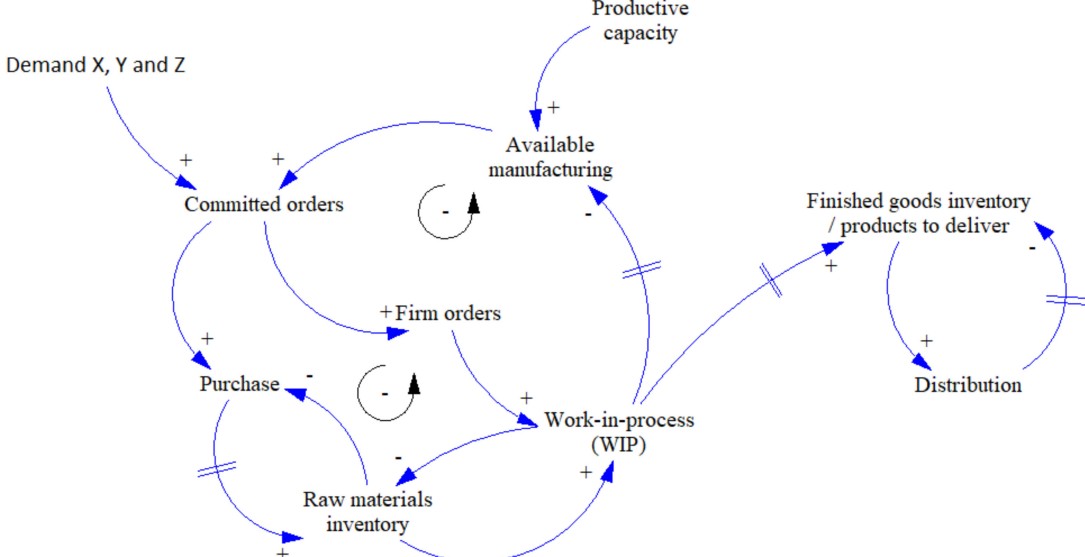

**Figure 4.** Influence diagram. Source: own elaboration. Designed in Vensim software.

The diagram shows the variables connected by arrows representing the relationship between them. The direction of these arrows corresponds to the direction of the effect. The "+" or "−" signs are the signs of the effect, where "+" indicates that the variables change in the same direction while "−" indicates the opposite.

The diagram represents an order-based supply chain model that starts with the top left corner with the exogenous variable Demand x, y, and z and continues with the committed orders for production and demand satisfaction. This graph can be understood in two hemispheres. The left hemisphere presents the two control loops: available manufacturing

and raw material inventory, while the right hemisphere relates to the distribution of products, and no loops are evident.

Regarding the left hemisphere, the first cycle comprises the committed products that correspond to the products that the supply chain will serve. These generate *Purchases* that are the amount of raw material necessary to produce the committed products, which varies depending on the order. Once this purchase is calculated, the *Raw Material Inventory* is created, which is the amount of raw material stored to be dispatched. The latter then transitions to *Work-in-process (WIP)* and affects the *Available Manufacturing*, which conditions the order approval policy. The second cycle, which is responsible for regulating production based on raw material inventories. It starts with the *Committed Orders* that define the *Firm Orders*, i.e., those accepted waiting for the purchase of raw material. Once the purchase is received, they become *Work-in-process (WIP)*, varying the *Available Manufacturing*. These cycles are stable and allow defining the order approval policy and the purchase of the raw materials.

On the other hand, the right hemisphere includes the *Products to Deliver*, which are stored in the *Finished Goods Inventory.* By definition, the finished product inventory does not increase or decrease the production capacity or the work-in-process inventory. Finally, they are distributed, generating a stability cycle on the system.

As mentioned before, these dynamic interactions between variables influence the responsiveness of supply chains, particularly in order management. This situation is verified through the flow and level diagram and its subsequent simulation.

*4.3. Model Variables*

The structure of an SD model contains stock and flow variables [11]. Taking as a starting point the conventions used in SD methodologies [40] and the variables related to SC management found in the literature review, the six elements included in the model are grouped and presented below:

- **Model parameters:** The parameters correspond to elements of the model that are independent of the system or its own constant that does not vary during the simulation [40], where they are found:
  1. Raw_Mat_Inv: the amount of raw material, which corresponds to the raw material consumption for each product demanded. The consumption is different for each product.
  2. Print_Load: capacity to load products per printer and machine, i.e., how many products a machine/printer can produce/print simultaneously.
  3. Printers: number of printers/machines installed in the system.
  4. PO_A_T: production order acceptance time.
  5. Dist_A_T: time of acceptance of the product to be distributed.
  6. Time_OA: time of acceptance of the product entry.
  7. Dist_Lead_Time: time it takes to deliver the demanded product to the customer varies according to the shipping characteristics (transport used and distance traveled).
  8. Print_Time: time to process an order.
  9. S_Lead_Time: supplier processing time.

- **Level variables:** The level variables or state variables represent the accumulation of flows, which in this case are stable; i.e., as they grow, they also leave the system.
  1. Committed_Orders: the products to be handled in the supply chain.
  2. Firm_Orders: the products accepted for production and awaiting raw materials.
  3. Raw_Mat_Inv: Raw Material Inventory corresponds to the amount of raw material that the supplier stores and is available for production.
  4. Work_In_Process: the products in the production process. Their behavior and development depend on the processing time.
  5. Finished_O_Inv: finished orders inventory refers to the number of finished units ready for delivery.

6. Delivery_Request: products that are accumulated in the finished goods inventory and are pending delivery.
7. O_Records: history of delivered products. It is used to verify product delivery and evaluate the model from an input–output perspective, where everything that the SC commits to doing is delivered.

- **Flow elements:** Flow elements that are understood as the variation of a level, representing changes in the state of the system:
  1. CO_Policy: acceptance policy corresponds to the products' units accepted to enter the supply chain and be produced and delivered.
  2. Inputs_Order: product entry is defined by the total load capacity or final stock, which defines the production batch to be accepted.
  3. Purchase: amount of raw material needed to produce all the orders; the consumption changes according to the type of demand to be produced.
  4. Prod_Order_E: production order entry. It corresponds to those orders that are taken into account according to the distribution network's operating policies. The availability of capacity and the existence of raw material for production are verified.
  5. O_Accepted_AS: products accepted according to the production orders' requirement and the stock of raw material to be produced.
  6. Raw_Mat_Exit: quantity of raw material sent by the supplier according to the producer's need. The number of kilograms/grams (unit of measure) needed to produce a product is considered.
  7. Order completed: final stage of the production process. From now on, the product will be considered as already elaborated.
  8. Product_Delivery: delivery of finished goods to the distributor.
  9. Delivered: products that leave the system and are considered delivered to the final customer.

- **Delays:** Delays are elements that simulate delays in transmitting information or material between system elements. All model delays are of infinite order since they produce an output equal to the input after a particular time. That is to say, the delay manifests itself in the output with the same input that arrived sometime before.
  1. Purchase Delay: represents the processing time of the supplier in producing and obtaining the raw material necessary for the production carried out by the manufacturers or assemblers.
  2. WIP: represents the sum of the products that are in process. Assuming that no product fractions are received, and no product fractions leave the System.
  3. Delivery Delay: represents the time it takes the distributor to deliver the product to the customer.

- **Auxiliary Variables:** Auxiliary variables are quantities with some significance to the modeler and with an immediate response time. They have been divided into base model auxiliary variables and structural policy auxiliary variables.

  Base model auxiliary variables:
  1. Productive_Capacity: total load capacity of the system, i.e., how many products can be produced simultaneously.
  2. Available_Manufacturing: available capacity or load availability; corresponds to the total amount of products that can go into production (of any given demand).

  Structural policy auxiliary variables:
  1. RM_Stock1: raw material stock after manufacturing the batch from demand 1.
  2. RM_Stock2: raw material stock after manufacturing the batch from demand 2. That is to say, the raw material stock available for demand 3.

3. F_Order1: number of products of demand one that can be manufactured according to the products that are for production and the raw material stock available at the moment.
4. F_Order2: number of products of demand two that can be manufactured according to the products that are for production and the raw material stock available at the moment.
5. F_Order3: number of products of demand three that can be manufactured according to the products that are for production and the raw material stock available at the moment. This step comes after meeting the first two demands.
6. FO_available_inv: products that can be manufactured based on the raw material stock available at the moment.

Approval policy based on manufacturing availability

1. A_Load1: products available after meeting demand 1.
2. A_Load2: products available after meeting demand 2. What is left from this part will be used to manufacture demand 3. This is the number of products that can be served from demand 1. For this step, it is necessary to analyze the product availability and the demand requirements.
3. Order1: number of products that can be served from demand 1. For this step, it is necessary to analyze the product availability and the demand requirements.
4. Order2: number of products that can be served from demand 2. For this step, it is necessary to analyze the product availability and the demand requirements. It also comes after meeting demand 1.
5. Order3: number of products that can be served from demand 3. For this step, it is necessary to analyze the product availability and the demand requirements. This measure comes after meeting the first two demands.
6. Order_Prodt: production order: quantity of each product delivered for production, previously having guaranteed the raw material stock and the production capacity availability.

- Exogenous variables: Exogenous variables are those whose evolution is independent of the rest of the system. They represent the actions of the environment upon it.

  1. Prodt_X: Products X. Represents the demand X in different periods.
  2. Prodt_Y: Products Y. Represents the demand Y in different periods.
  3. Prodt_Z: Products Z. Represents the demand Z. in different periods.

### 4.4. Data-Flow Diagram

Based on the influence diagram and the definition of model variables, the data flow diagram was constructed using the conventions indicated in Section 4.2, summarized in Table 3.

The model is structured in four sectors, each one related to the defined supply chain links: order display, supplier, manufacturers, and distribution network, alongside the SC order traceability and the operating policies for inventory and availability. Figure 5 presents the data flow diagram that depicts the multi-product supply chain's behavior, taking into account the definition of the order approval policies and the purchasing policy. This representation corresponds to a simplified approximation to the supply chain and its order management. It also operates in vector graphics except for the supplier link because it only recreates a single raw material which is used for different products.

**Table 3.** Elements of the model.

| Element | Name | Description |
|---|---|---|
| | Parameter | Constant value of the system that does not change during the simulation. |
| | Level variable (Stock) | Corresponds to the state variables in systems theory and represents the flows accumulation. |
| | Flow variable (valve) | Defines the behavior of the system. |
| | Delay | Simulates delays in material or information transmission between elements of the system. |
| | Table | Represents a nonlinear relationship between two variables. |
| | Auxiliary variable | Element that has certain meaning or interpretation for the modeling with an immediate response. |
| | Exogenous variable | Has an independent evolution from the system evolution. It represents an interaction of the system with the exterior. |
| | Information channel | The transmission of information that does not require storage. |

Taken from: [40].

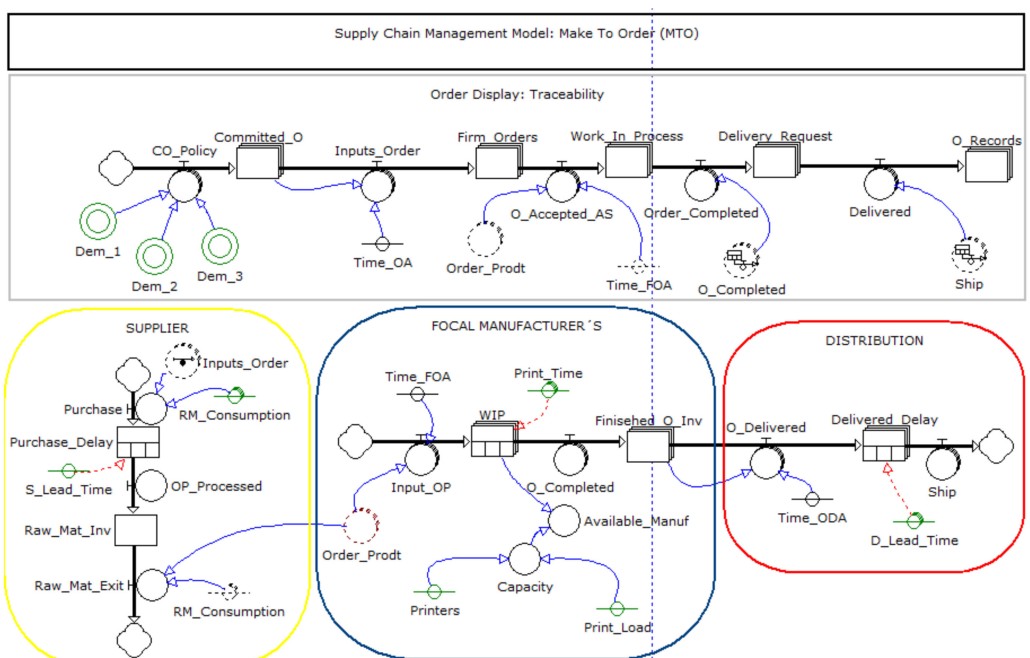

**Figure 5.** Data flow diagram. Order-based SCM (MTO). Source: own elaboration. Evolution software.

The vector graphics have the advantage of representing, simultaneously, different operating conditions. In the case of the proposed model, it allows representing the behavior throughout the supply chain with three types of product 1(X), 2(Y), and 3(Z), with different material consumption, processing times, and distribution times for each one.

The order management process is developed transversally to the SC, relationship with the three links in the chain. It starts with the multi-product demand (product 1, product 2, product 3), which are exogenous variables of the system. In this part, the products to be handled by the manufacturers are obtained. The order approval is determined based on the defined policies.

Once an order is accepted, it is communicated to the supplier, who manages the raw material inventory and dispatches the necessary quantities for production to the manufacturers. In this process, there is a purchase delay caused by the processing that the supplier must have in the acquisition of products or its own internal processing. The raw

material purchasing policy represented in the model is the immediate purchase of what is committed to being produced by the manufacturers.

Subsequently, the production order is generated according to the products that are in the firm orders status, which go to the processing stage according to the available manufacturing determined by the productive capacity. The latter is calculated with the number of products that a machine can produce and the number of machines/printers established in the system. Production times have been defined from the design stage, where raw material consumption and processing time (printing time) have been previously determined.

Once the entire production process has been carried out, it goes to the finished goods inventory of the focal manufacturers, who deliver the product to the distribution network. In this stage, a delivery delay appears, which varies according to the characteristics of the transport and routing system used by the distributor, which the schedules used by the modeler can define.

Moreover, the structural operation management policies of the model were designed to associate the approval policy based on stock and the approval policy based on manufacturing availability, as shown in Figure 6. These are policies that the focal manufacturers consider for prioritizing order processing and the generation of production orders.

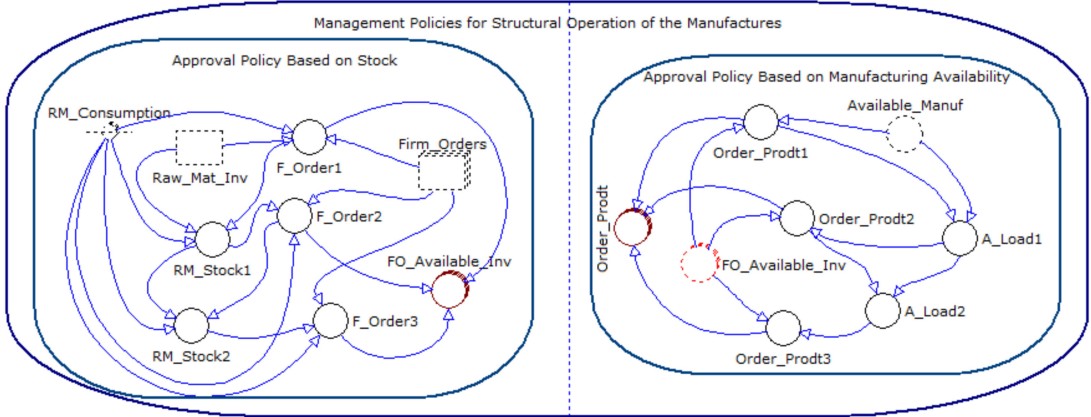

**Figure 6.** Structural operating policies of the model. Source: own elaboration. Evolution Software Evolution 4.6.

Consequently, the raw material inventory is analyzed to ensure that it can guarantee production. Afterward, the availability of the production capacity is reviewed. This element generates the order that is entering the process. Simultaneously, the priority criteria intervene to handle first product 1, then product 2, and, finally, product 3.

The latter means that the *Firm Orders* are a priority for the manufacturers, and they will continue producing them until there is no more demand. When they finish with product 1, they will continue with product 2 and successively with product 3.

### 4.5. Equations

In addition to the status equations that can be seen directly in the data flow diagram, Table 4 shows the equations for each of the sectors.

**Table 4.** Mathematical equations in data flow diagram.

**Order display: Traceability**

CO_Policy:Flow_
Definition = [Dem_1,Dem_2,Dem_3]
Description = Approval policy is to produce all required products.
Handling by order: (Dem_1, Dem_2 y Dem_3)
Inputs_Order:Flow_
Definition = [(Committed_O [1]/Time_OA),(Committed_O[2]/Time_OA),(Committed_O[3]/Time_OA)]
Description = The product input is defined by the productive capacity (Variable) which defines the production batch to be accepted (current case batches of 10).
O_Accepted_AS:Flow_
Definition = [(Order_Prodt[1]/Time_FOA),(Order_Prodt[2]/Time_FOA),(Order_Prodt[3]/Time_FOA)]
Description = According to the requirements of production orders and raw material stocks, these are the accepted products to be produced.
Order_Completed:Flow_
Definition = [O_Completed[1],O_Completed[2],O_Completed[3]]
Description = Processed product is the output or finished good.

**Operating policies**

A_Load1:Auxiliary_
Definition = (Available_Manuf-Order_Prodt1)
Description = For order available inventory after meeting Dem1.
A_Load2:Auxiliary_
Definition = (A_Load1-Order_Prodt2)
Description = For order available inventory after meeting Dem2. What is left will be assigned to meet Dem3.
FO_Available_Inv:Auxiliary_
Definition = [F_Order1,F_Order2,F_Order3]     Description = Taking into account what I want to attend to, these are the possible ones to be produced with the available raw material.
F_Order1 :Auxiliary_
Definition = INT(Min(Firm_Orders[1],(Raw_Mat_Inv/RM_Consumption[1])))
Description = The quantity of Dem_1 products that can be produced according to the products and raw material available for production.
F_Order2:Auxiliary_
Definition = INT(Min(Firm_Orders[2],(RM_Stock1/RM_Consumption[2])))
Description = The quantity of Dem_2 products that can be produced according to the products and raw material available for production after meeting Dem_1.
F_Order3:Auxiliary_
Definition = INT(Min(Firm_Orders[3],(RM_Stock2/RM_Consumption[3])))
Description = The quantity of Dem_3 products that can be produced according to the products and raw material available for production after meeting the first two demands.
Order_Prodt:Auxiliary_
Definition = [Order_Prodt1,Order_Prodt2,Order_Prodt3]
Description = Production order: Quantity of each product delivered for production, having previously guaranteed the availability of raw materials and the available production capacity.
Order_Prodt1:Auxiliary_
Definition = Min(FO_Available_Inv[1],Available_Manuf)
Description = quantity of products that can be handled from Dem1, according to the FO available inventory and the product requirements.
Order_Prodt2:Auxiliary_
Definition = (Min(FO_Available_Inv[2],A_Load1))
Description = quantity of products that can be handled from Dem2, according to the FO available inventory and the product requirements after meeting Dem_1.
Order_Prodt3:Auxiliary_
Definition = (Min(FO_Available_Inv[3],A_Load2))
Description = quantity of products that can be handled from Dem3, according to the FO available inventory and the product requirements after meeting the first two demands.
RM_Stock1:Auxiliary_
Definition = (Raw_Mat_Inv-(F_Order1*RM_Consumption[1]))
Description = Raw material inventory after Dem_1.
RM_Stock2:Auxiliary_
Definition = (RM_Stock1-(F_Order2*RM_Consumption[2]))
Description = Raw material inventory after Dem_2. What is left will be assigned to handle Dem_3.

**Table 4.** *Cont.*

| Supply Chain |
|---|

| *Supplier* |
|---|

Purchase:Flow_
Definition = (Inputs_Order[1]*RM_Consumption[1]) + (Inputs_Order[2]*RM_Consumption[2]) + (Inputs_Order[3]*RM_Consumption[3])
Description = The amount of raw material needed to produce all the committed orders, which consumption varies according to the type of demand to be produced.
Raw_Mat_Exit:Flow_
Definition = (Order_Prodt[1]*RM_Consumption[1]) + (Order_Prodt[2]*RM_Consumption[2]) + (Order_Prodt[3]*RM_Consumption[3])
Description = The amount of raw material sent by the supplier according to the producer's needs. This measure is calculated by kilograms.

| *Focal manufacturer* |
|---|

Available_Manuf:Auxiliary_
Definition = Capacity-(SUMARETARDO(WIP[1]) + SUMARETARDO(WIP[2])+SUMARETARDO(WIP[3]))
Description = For order available inventory: Total quantity of products that can enter into production (of any demand).
Capacity :Auxiliary_
Definition = Print_Load*Printers
Description = System productive capacity: Number of products that can be handled simultaneously.
Input_OP:Flow_
Definition = [(Order_Prodt[1]/Time_FOA),(Order_Prodt[2]/Time_FOA),(Order_Prodt[3]/Time_FOA)]
Description = Entry of the production order, taken into account according to the operating policies of the local manufacturer, which verifies the FO and raw material available inventory for production.

| *Distribution Network* |
|---|

Delivered:Flow_
Definition = [Ship[1],Ship[2],Ship[3]]
Description = These are the products that leave the system and are considered delivered to the final customer.
O_Delivered:Flow_
Definition = Finished_O_Inv/Time_ODA
Description = Product distribution order acceptance time.

Source: Own elaboration. Evolution Software.

*4.6. Model Validation*

Once the model was formulated, it was validated by performing the analysis proposed by [10] since it allows questioning the structure, behavior, and policies that make up the model. It is also widely studied in the literature since it helps determine the system's stability when parameters and exogenous variables are modified.

With the structure validation, the relationships used in the system are judged in comparison with the actual processes of a supply chain. In this sense, it was necessary to apply extreme conditions tests to identify structural failures. Extreme values were attributed to the parameters and exogenous variables to verify their behavior at the end of the tests.

I Source: own elaboration. Evolution Software. In the first test, it was assumed that there was no demand in the supply chain, which, at the same time, would mean that all the system states would remain at zero since the initial conditions corresponded to this value. This behavior can be seen in Figure 7.

In contrast to the extreme conditions test, some other tests were applied where the demand was modified by oversizing each of the products' quantity. Consequently, the values of the parameters purchase time, printing time, and distribution time were increased with the same extreme behavior compared to a natural system. In this way, it was verified that everything demanded leaves the system through the *Order Records* variable (O_Records), as shown in Figure 8.

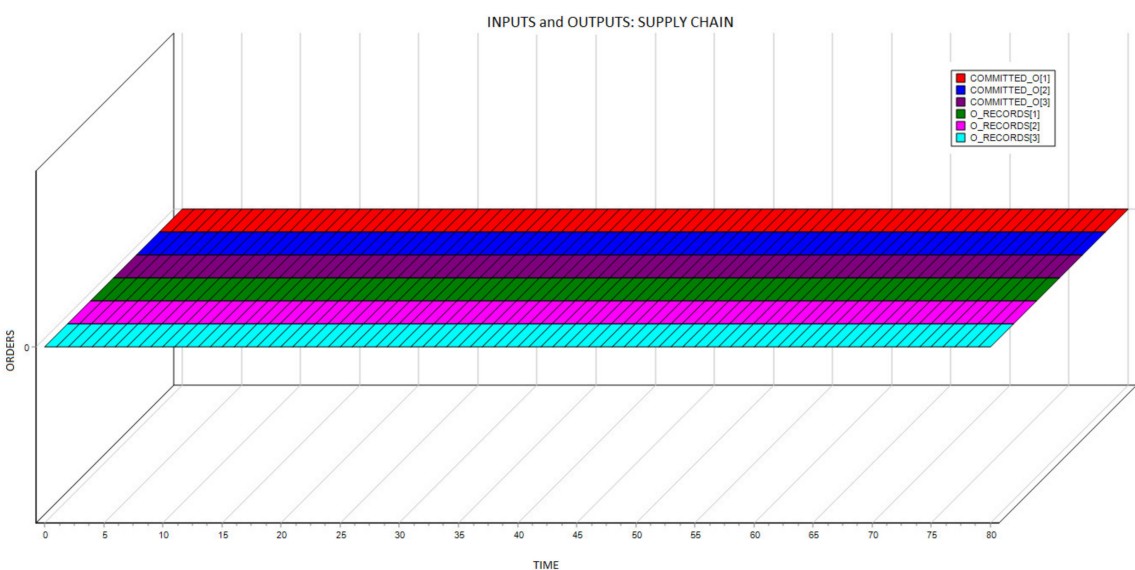

**Figure 7.** Extreme conditions test. Source: own elaboration. Evolution Software.

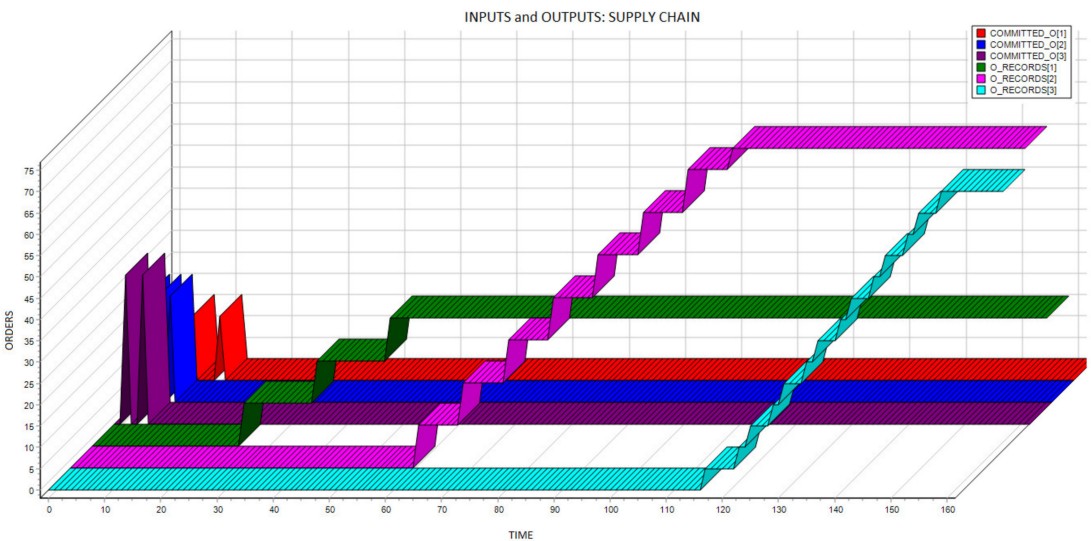

**Figure 8.** Validation test with oversized demand. Source: own elaboration. Evolution software.

On the other hand, another model evaluation was carried out to check the system's behavior through a sensitivity test. The impact of the change of parameters that were considered highly sensitive for the production system was observed, in this case those that affect the capacity of the system, i.e., the productive capacity for the focal manufacturers, which is conditioned by the number of printers/machines and the total load of products handled by each printer/machine. Figure 9 shows the auxiliary variable "Available manufacturing" over time, with a constant productive capacity, $1\times$, $5\times$, and $15\times$.

Thus, the productive moment with demand is visualized, responding to the requested demand, as well as the non-productive moment that shows the installed capacity. This result is consistent with the expected logical evidence.

From the tests performed on the model, it was possible to conclude that the representations obtained from the behavior of the supply and distribution system of a focal manufacturer correspond to those referenced as a supply chain system, taking into account that the variables contemplated are associated with the management processes of a company.

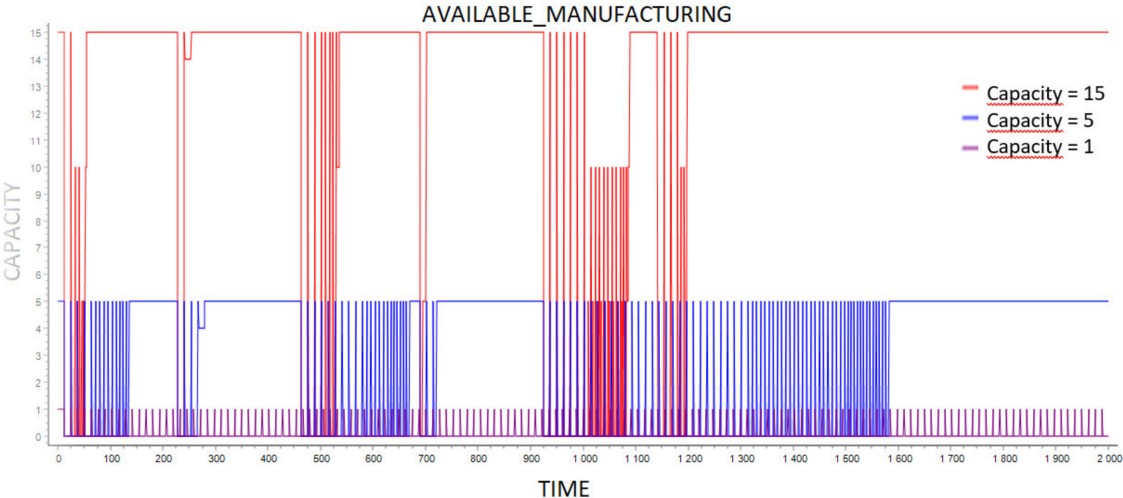

**Figure 9.** System sensitivity test. Source: own elaboration. Evolution software.

Since additive manufacturing in the manufacturing area is still in its infancy [44], there is no accurate data on the practice in natural environments. For this reason, simplified data of existing standard cases are proposed with assumptions that allow evaluating and concluding if it presents a feasible behavior within the expected, without presenting dramatic alterations.

### 4.7. Model Sensitivity Analysis

The initial conditions and model parameters were defined based on the exogenous variable "demand" to perform the model sensitivity analysis. The focal manufacturer has three types of demands to meet, as shown in Table 5. These values are independent for each product, correspond to a period of six months, and are kept the same for the traditional and additive approach so as to allow comparison.

**Table 5.** Model Conditions Demands.

| Month | Hour | Dem1 | Dem2 | Dem3 |
|---|---|---|---|---|
| 1 | 1 | 20 | 30 | 20 |
| 2 | 217 | 16 | 0 | 0 |
| 3 | 452 | 45 | 30 | 35 |
| 4 | 678 | 10 | 0 | 5 |
| 5 | 913 | 100 | 115 | 35 |
| 6 | 1130 | 50 | 0 | 35 |
| | TOTAL | 241 | 175 | 130 |

Source: own elaboration.

The unit of time selected for the model was hours, so it was also necessary to determine the hour at which the demand was updated. The third, fourth, and fifth columns correspond to the demand that exists at that time for each type of product. For example, in hour 217, there is a demand for 16 products for Dem1, 0 products for Dem2, and 0 products for Dem3.

After establishing the demand, it was necessary to define the parameters for the behavioral simulation, as presented in Table 6, with the values for traditional manufacturing (TM) and additive manufacturing (AM).

**Table 6.** Model parameters.

| Parameters | TM | AM |
|---|---|---|
| Machine/Product | (4:1) | (1:4) |
| Processing_T_dem1 (Hours) | 1 | 3 |
| Processing_T_dem2 (Hours) | 2 | 4 |
| Processing_T_dem3 (Hours) | 4 | 6 |
| RM_Consumption_Dem1 | 11 | 9 |
| RM_Consumption_Dem2 | 2 | 1 |
| RM_Consumption_Dem3 | 8 | 7 |
| S_Lead_Time (Hours) | 4 | 4 |
| D_Lead_Time_Dem1 (Hours) | 12 | 12 |
| D_Lead_Time_Dem2 (Hours) | 12 | 12 |
| D_Lead_Time_Dem3 (Hours) | 12 | 12 |

Source: own elaboration.

For the machine–product relationship, by definition, the manufacture of a product through TM features four machines intervening, each one producing 25% of the final part; while in AM, a single printer is capable of producing four products since it works them as a single unit [45].

Regarding the processing time, a different value in hours is defined for each type of demand, taking into account the complexity of the design and geometry required, and the percentage increase is projected for each one of them, considering that printing requires more individual production time [46,47].

Regarding raw materials, each TM demand had a different consumption assigned, whereas for AM consumption, researchers referred to some documented cases where the material usage decreases by at least 10% compared to the traditional [44,48]. Based on the latter, the approximations were made as shown in the table above.

Finally, it is proposed that the distribution time is the same no matter the type of demand or manufacturing approach adopted since the origin–destination path is the same in this case.

The parameters mentioned above may be subject to variations by the modeler if he/she wishes to modify proposed times or quantities. The latter allows the model to be used in different application industries due to its adaptability to different environments.

With the conditions of the model defined, the simulation was carried out under the two manufacturing approaches. The first result shows the TM and AM raw material consumption, as shown in Figure 10.

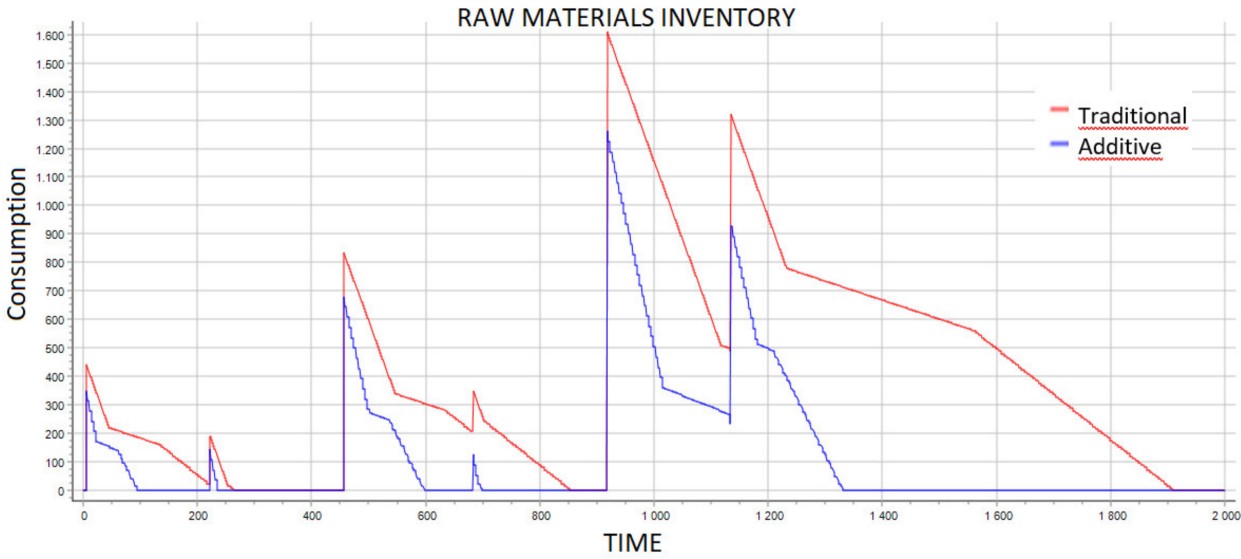

**Figure 10.** Raw material behavior. Source: own elaboration. Evolution software.

For the additive case (blue), it is observed that smaller quantities of raw material are required at the time of request compared to the traditional case (red). The previous fact demonstrates that its consumption is quicker given that there is faster production time, as is evident in the results. This means that the supplier must adopt agile inventory management due to a quicker order approval by the manufacturers.

Another interesting result is how the system available manufacturing responds to the demand for the requested products and approval of new ones, if necessary. Figure 11 represents production over time.

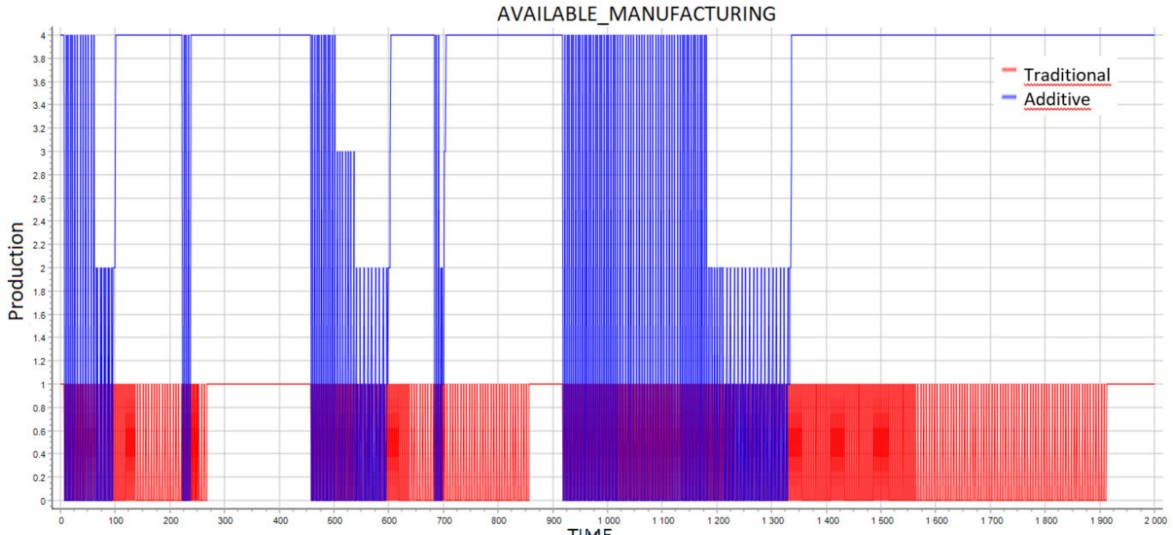

**Figure 11.** Production over time. Source: own elaboration. Evolution software.

As defined in the parameters, the traditional SC needs several machines to produce a single product. The model states that four devices are used to make a single product, which means a 0.25 ratio for each machine, whereas for additive SC, one printer can handle several products. The model states that one single printer is capable of producing four products at the same time.

This situation generates higher available manufacturing for the additive case since faster production remains independent from the production time per product. When reviewing the machinery utilization index, the traditional SC is higher than the additive SC because it requires more activity to generate the same result.

On the other hand, the order records for each of the products were plotted. Figure 12 represents the order record for demand 1. Considering that product 1 has the highest demand of the three products, a prolonged upward behavior can be observed, where demand satisfaction is reached more quickly in the additive approach.

In product 2, the result of the first product is maintained, i.e., the ASC satisfies demand more quickly. Figure 13 shows the behavior, where the valleys correspond to periods where demand has been satisfied, and the company is waiting for a new order to continue production. As more valleys are generated in the ASC, it is understood that the lead time is shorter because there is an immediate time response.

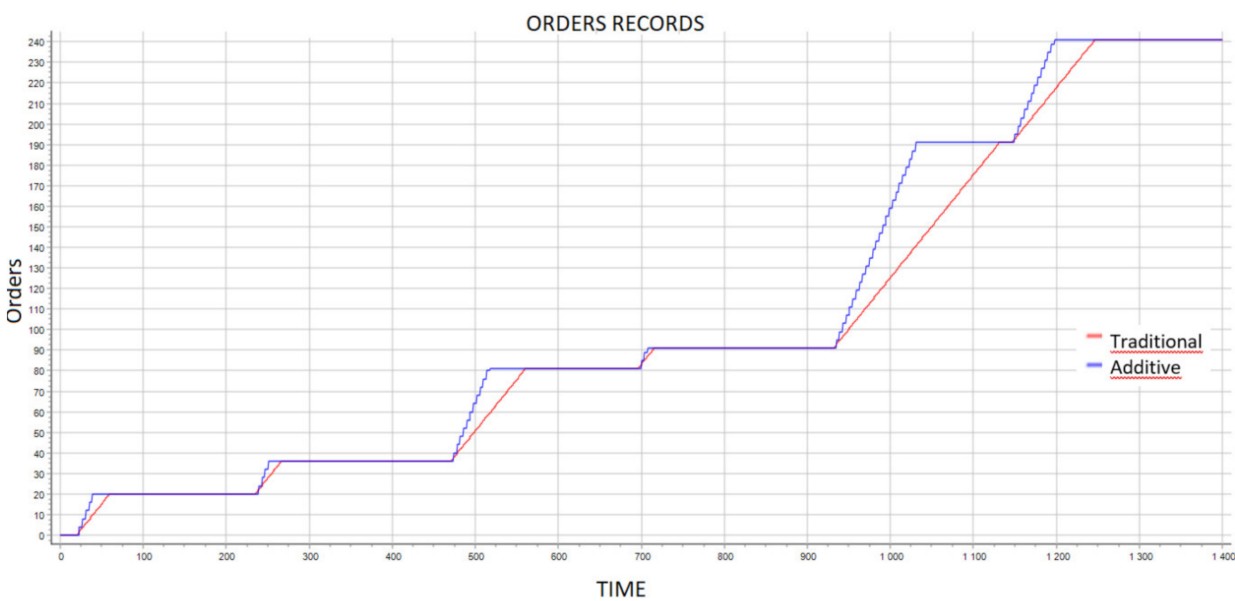

**Figure 12.** Order record of Demand 1. Source: own elaboration. Evolution software.

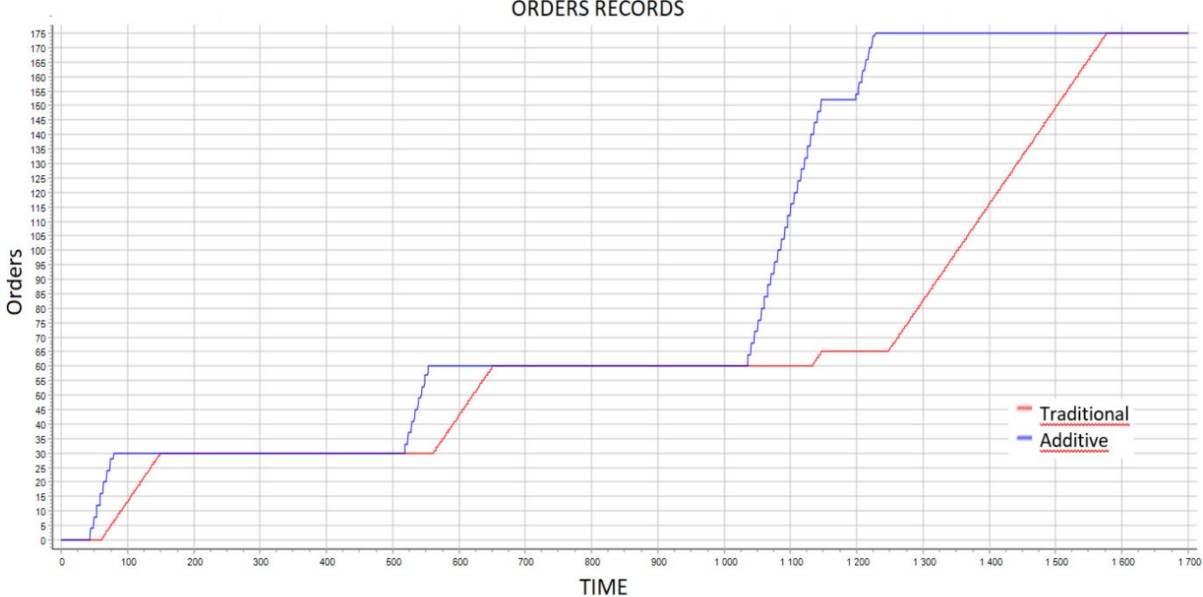

**Figure 13.** Order record of Demand 2. Source: own elaboration. Evolution software.

Moreover, Figure 14 shows the behavior of product 3. Although the demand is similar to product 2, the response times and the raw material consumption are different, which generates changes in the graph, but the result remains the same. The ASC responds more quickly because the production time is faster than the TSC. Comparing the three graphs in the lower quantity demands, it is easily evident that the ASC production time (Lead) is shorter.

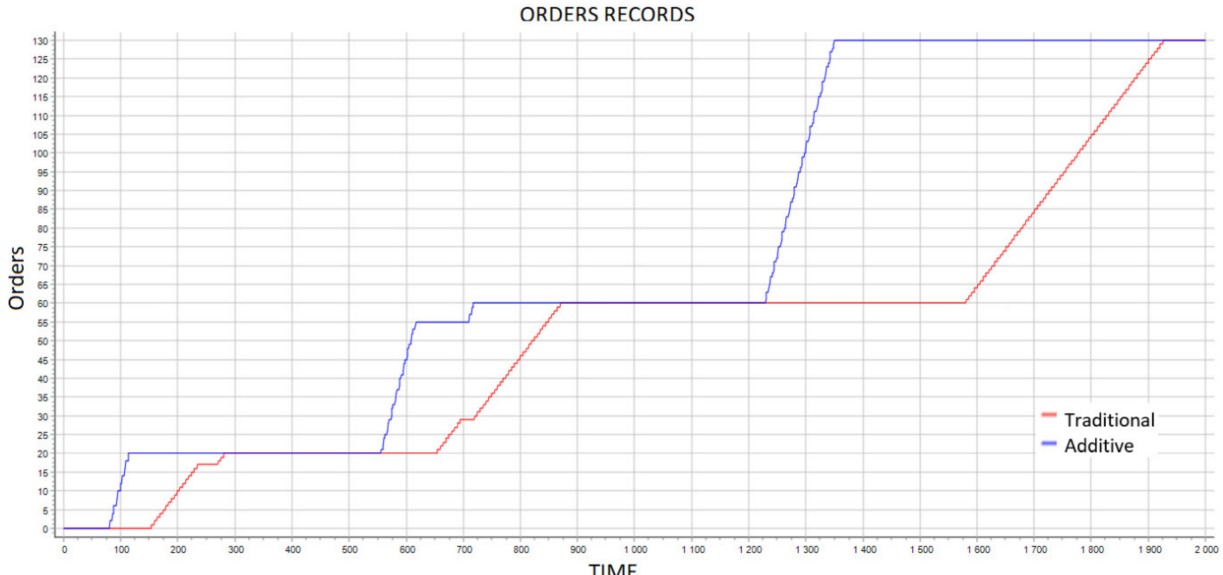

**Figure 14.** Order record of Demand 3. Source: Own elaboration. Evolution software.

When analyzing the order records as a whole, it is possible to note that the ASC excels in the speed of delivery. As demand increases, the advantage decreases, i.e., additive manufacturing positively impacts response times, especially in small production batches. To illustrate more precisely the comparison between the TSC and the ASC response times, Table 7 was constructed to identify the demands of each product for the simulation time and the delivery time after distribution.

**Table 7.** Order completion time.

| Delivery (Hours) Dem 1 | | | Delivery (Hours) Dem 2 | | | Delivery (Hours) Dem 3 | | |
|---|---|---|---|---|---|---|---|---|
| **Dem1** | **TSC** | **ASC** | **Dem1** | **TSC** | **ASC** | **Dem1** | **TSC** | **ASC** |
| **20** | 59 | 39 | **20** | 59 | 39 | **20** | 59 | 39 |
| **16** | 266 | 251 | **16** | 266 | 251 | **16** | 266 | 251 |
| **45** | 560 | 518 | **45** | 560 | 518 | **45** | 560 | 518 |
| **10** | 715 | 708 | **10** | 715 | 708 | **10** | 715 | 708 |
| **100** | 1131 | 1031 | **100** | 1131 | 1031 | **100** | 1131 | 1031 |
| **50** | 1246 | 1198 | **50** | 1246 | 1198 | **50** | 1246 | 1198 |

Source: own elaboration. Evolution software.

These are some of the main differences gathered from the order completion time analysis performed on the ASC and TSC:

*Demand 1:*

— The response was about 33.9% faster in the ASC than in the TSA. The following advantage ranged between 1% and 8.8%.
— There was a difference of 3.9% for the total compliance of the 241 products.

*Demand 2:*

— The behavior in demand 2 remained the same. The difference was 47% in the first demand, followed by 14.8%, finishing the 175 products 22.5% faster than the TSC.

*Demand 3:*

— The behavior patterns of demand 3 were much higher since in the first demand, it reached 59.4%, and the other requirements range between 17.5% and 29.9%.

As mentioned before, through the analysis of the total closure time, it is possible to notice better performance with additive manufacturing than the traditional approach, especially with small batches.

## 5. Discussion

The contribution of this research is to recreate a starting point (base model) through a simplified supply chain structure that will allow analyzing the behavior of industrial transformation with the appropriation of disruptive technologies, in this specific case, the analysis of the variables that affect additive manufacturing. In this sense, the model will be an experimental laboratory to study what could happen with the behavior of the network structure, generating new questions such as: What would happen if there were more actors in the supply chain? What would happen if additive production was centralized or distributed in different regions? What would happen with inventory levels and transportation times? What would happen if the structures of the chains were changed?

Likewise, the model could include variables of processing costs, transportation costs, and printing costs and could visualize the economic impacts that the emergence of additive technology may have. Moreover, it could measure the environmental impacts on the structure, evaluate the supply chain behavior's optimization, and define the best response to production problems.

The model was based on systemic thinking, applying the system dynamics methodology to include the totality of the SC management processes, representing all the elements that make it up (suppliers, focal manufacturer, and distribution network): the purchasing, production, and product delivery processes to the final customer, representing the order management and the interaction of the actors. The model offers the opportunity to visualize the behavior of a complex system, such as the supply chain, allowing managers to obtain a degree of confidence through simulation, facilitating the planning of future and experimental environments.

Finally, the model allows comparing additive and subtractive behavior to contrast the simulation prediction with reality in the presentation of different scenarios. By modifying the variables, it will be possible to visualize emerging characteristics of the supply chain with "better" behaviors, savings in processing times, distribution times, and inventory levels. The advantage of approaching the model through system dynamics is that it represents cycles that can be formulated as a set of nonlinear differential equations, i.e., there is no fixed solution; instead, there is an infinite number of possible solutions to the behavior of the SC, facilitating the visualization of these characteristics and the easy-to-understand representation of a complex system.

The model has limitations in the inclusion of complex variables, such as the level of design, product customization, or the visualization of the change in information and knowledge management (patents), which impact the new business models appropriated by technology. In addition, it should be considered that the model does not provide specific optimization data since, as mentioned above, it is a starting point for considering multiple alternatives and future scenarios. Likewise, if researchers want to improve the reliability of the results, it is fundamental to base their investigation on real case studies because of the SC's complexity.

In the present study, aspects such as the decrease of machinery in the supply chain and the level of knowledge of the new human resources are not considered; therefore, it is necessary to explore the influence of new variables.

## 6. Conclusions

This paper reaffirms the importance of system dynamics to represent the behavior of the supply chain, visualizing the links that integrate it (supplier–focal manufacturer–distribution network). It also allows for appreciating the role of the delays presented in each link. The system influence diagram recreates the SCM as a whole, and the data flow

diagram recreates the relationships of the production process, allowing us to consider the impact of additive technology.

The vector model made it possible to represent the mathematical complexity of a set of differential equations, the smallest representation with 26 and the proposed scenario with 66, described in terms of flows, levels, and delays, allowing the reader to analyze the supply chain as a study phenomenon in a simple way. It also describes the particularity in order management (MTO) as the variation of the demand requirement and the product characteristics regarding raw material consumption, printing times, and distribution times.

In this way, a Make to Order (MTO) supply chain management was represented in a simplified form, allowing to contemplate characteristics of the AM and the TM, such as the transformation process, raw material management, inventory purchase, and product distribution. The sensitivity analysis performed on the AM and MT confirmed that the AM presents shorter lead times in the SC, higher production capacity, and lower raw material inventory levels.

Some of the potential impacts of the MA on the chain management processes were supported. The latter allows deducing high levels of customization, greater control of order traceability, lower storage levels, and less material transportation, reflected in lower costs and time.

This model constitutes a starting point to consider different alternatives for the functioning of the supply chain, which contemplates structural operating policies and operating policies in terms of process management. Furthermore, this phenomenon demands future research to formulate models that could facilitate the recreation of impact scenarios in appropriating technology, determining centralized and distributed supply chains, different roles of the elements, changes in the links, and the cost-benefit implications that this may represent.

**Supplementary Materials:** The following are available online at https://www.mdpi.com/article/10.3390/pr9060982/s1, Table S1: Applications—System Dynamics Modeling in Supply Chain Management; Table S2: Study variables in system dynamics models based on the Supply Chain.

**Author Contributions:** Conceptualization, J.N.R. and S.M.V.A.; methodology, J.N.R. and H.H.A.S.; software, J.N.R. and H.H.A.S.; validation, J.N.R. and H.H.A.S.; formal analysis, J.N.R. and H.H.A.S.; investigation J.N.R. and A.O.; resources, J.N.R. and A.O.; data curation, J.N.R. and H.H.A.S.; writing, review and editing, J.N.R., H.H.A.S., A.O. and S.M.V.A.; visualization J.N.R.; supervision A.O. and H.H.A.S.; project administration, J.N.R. and A.O. All authors have read and agreed to the published version of the manuscript.

**Funding:** This research received no external funding.

**Institutional Review Board Statement:** Not applicable.

**Informed Consent Statement:** Not applicable.

**Data Availability Statement:** All data are available upon request.

**Conflicts of Interest:** The authors declare no conflict of interest.

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
