# Peer review of "System Dynamics Modeling in Additive Manufacturing Supply Chain Management"

_processes, doi:10.3390/pr9060982_

Round 1

Reviewer 1 Report

The reviewed paper proposes an interesting problem, which has theoretical and also practical importance. Presented research is focus on supply chain, additive manufacturing and system dynamics. The main goal of reviewed paper is to reaffirm the importance of system dynamics to represent the behavior of the supply chain, visualizing the links that integrate it. Author(s) considered a vector model to represent the mathematical complexity of a set of differential equations, the smallest representation with 26 and the proposed scenario with 66, described in terms of flows, levels, and delays. Author(s), also stated, that presented model constitutes a starting point to consider different alternatives for the functioning of the supply chain, which contemplates structural operating policies and operating policies in terms of process management. Finally, author(s) observed, that there is a need for formulation of models that could facilitate the recreation of impact scenarios in appropriating technology, determining centralized and distributed supply chains, different roles of the elements, changes in the links, and the cost-benefit implications that this may represent. Author(s) could clearly explain what is the breakthrough in this paper.

The manuscript’s weaknesses are:

  1. I suggest author(s) add precisely formulated advances of presented research and results.
  2. Lack of some important published related works, which should be considered and included in the ”References” section.
  3. It is confusing and not cleary defined which part of this research belongs to authors of this paper only, since author(s) have not included sources, below each table, figure and algorithm.
  4. Lack of more extended discussion about practical approaches of obtained results.

Paper requires revision before publication.

A point-by-point list of constructive recommendations for the improvement of the manuscript are presented below:

  1. Authors should clearly explain what is the breakthrough in this paper.
  2. A comparative analysis can be provided to make a comparison between this study and other relevant works.
  3. I also suggest author(s) should add discussion about pros and cons of considered problem to clearly identify the benefits.
  4. Author(s) should define, what is new and uniq compared to other published papers about the same research topic.
  5. I suggest update abstract to highlight most important findings of this research.
  6. It would be interesting for readers if the paper include theoretical section about extended number of different potential practical applications in different areas, if possible.
  7. At least, these two important works should be integrated in the references: 10.1080/00207543.2018.1488086, 10.1108/S0276-897620200000020003.
  8. Below figures and tables author(s) should add source.
  9. Keywords should be in English, not in Spanish.
  10. Additionally, the text should be checked for spelling and grammar before publication. Paper contains some amount of typos that need to be corrected throughout the paper. There are several minor language errors in the text.

Author Response

Good morning, please see the attachment file

Reviewer 2 Report

Dear authors,

thank you for giving me the possibility to review :"System Dynamics Modeling in Additive Manufacturing Supply Chain Management ". I hope you and your family are safe and well.

Looking at your paper I find it interesting but you have to improve literature and something more.

About literature I suggest to improve it

Introduction :
Please write more about Supply Chain management and System Dynamics.
I think you have to better improve introduction.
About methods : the use of System Dynamics is largely used. I think you have to dedicate a general brief section to it. At this regard I suggest to start from "System Thinking" book  and then develop more. At this regard I suggest one paper  https://doi.org/10.15866/iremos.v9i4.9688

These authors use SD in different field. It can be a start to create a section and where you can use the "System Thinking".

The paper is good, but I find it with lack in the fields I suggest to improve. Of course my suggestions are the start, then you can choose other papers to improve your literature.

Do these changes, I'll sure you'll get the goal

Author Response

Good Morning, 

Please see the attachment, the changes are in pruple color. 

Reviewer 3 Report

Dear authors,

I read your paper and I find it very interesting. I'll suggest some improvements  in organization/literature.

The use of System Dynamics si interesting and studied. At this regard I suggest to improve your literature with the papers you would like to

They used System Dynamics in Safety Management but is interesting how they create the cause /effect relationship. Even if System Dynamics is well studied I think you have to better introduce it and profundize it (I'd create a little section only for it). I appreciate the first citation to Forrester, but we have to do something more...

For the rest the paper is quite well structured.

Author Response

Good morning, 

Please see the attachment, the response are in purple color. 

Thank you, 

Round 2

Reviewer 1 Report

The current version contains the required materials as suggested and it is acceptable for publication in the Journal.

I have no further suggestions regarding the improvements of the content.

Reviewer 2 Report

I congratulate with you. I say accept, but if you can, I suggest to improve the citations with the author I suggested (more papers).

Reviewer 3 Report

I congratulate with you. I say accept, but if you can, I suggest to improve the citations with the author I suggested (more papers).